# OpenReview forum: "Large Language Model Cascades with Mixture of Thought Representations for Cost-Efficient Reasoning"
_ICLR.cc/2024/Conference — ICLR 2024 poster_

### Official Review · Reviewer_6oKd · 2023-10-31

**Soundness:** 3 good
**Presentation:** 3 good
**Contribution:** 3 good
**Rating:** 6
**Confidence:** 3

**Summary:**

This paper introduces the notion of "model cascades" which is a method to offload easier problems to weaker models, which saves costs. They propose a simple method: answer consistency of the weaker LLM. Intuitively, this just means when the weaker llm is inconsistent, offload the task, because the model is uncertain. From there, the "stronger" llm performs inference on the task to solve it. They find this method that increase performance while decreasing cost by a significant margin.

**Strengths:**

- The idea is simple and understandable. It's also quite clear why this would help performance -- namely that in highly uncertain situations, samples from an LLM are likely to diverge and lead to overall increased entropy -- so contributing these cases to a stronger model is likely to lead to improved performance.
- The evaluation is extremely comprehensive. I appreciate the breadth of evaluation across different reasoning tasks
- The extended results in 3.6 are quite interesting as well -- clearly stating the limitations around how weak the weak llm can be is useful

**Weaknesses:**

In general, it's clearly stated throughout the paper that this method is aimed at "reasoning tasks" which indicated focus on datasets like gsm8k or big bench hard -- where the model must reason or understanding challenging problems. Nevertheless, I'm a bit concerned about how well this method would generalize to factuality based tasks or tasks that concern reasoning about facts/knowledge. In these situations it may be the case the model is highly confident (though it is incorrect) about a few pieces of knowledge which causes it to fail to reason correctly. Understanding that this paper is mostly about reasoning tasks, I'm still a bit concerned about how this method could be limited by the overconfidence in incorrect knowledge, and I believe it could be useful to evaluate this potential limitation to better inform readers about how this method may be useful.

**Questions:**

- For tasks that require a specific piece of knowledge are the ever situations where the weaker llm is confident, though incorrect, which causes the task not to be allocated to a more accurate & powerful model?

---

> ### Author Response · Authors · 2023-11-20
> **Thank you for your review**
>
> Thank you for your insightful feedback, especially your recognition of our comprehensive evaluation.
>
> Here are the responses to each of your questions:
>
> **W1/Q1: The weaker LLM may be overconfident in incorrect knowledge and fail to send the questions to the stronger LLM.**
>
> Thanks for your feedback! Yes, we acknowledge the inherent limitation of the LLM cascade method, wherein if the weaker LLM exhibits a high level of confidence in an incorrect factual answer, the pipeline would return that answer. Addressing this issue may necessitate additional measures such as external knowledge integration or human feedback, which is beyond the scope of this work. We have added this point in the limitation discussion of our revised paper and will consider it as an important future work.
>
>
>
> However, we note that our method is still effective on factual reasoning tasks. That is again owing to the fact that using different prompt representations could trigger different reasoning paths, which often results in more trustworthy answers when the two representations agree with each other. While completely addressing the overconfidence issue of LLMs is beyond our scope, we note that our idea of Mixture of Thought can indeed mitigate this issue.
>
>
>
> We have verified this idea in the newly added Appendix J based on the StrategyQA dataset. An example is shown in Figure 10. For the question "Is a curling iron necessary in curling?", the golden answer is "No, curling is an ice sport and doesn't need a curling iron". However, most of the CoT answers are "yes" with hallucinations about the concept of "curling". In contrast, most of the PoT answers are "No". The PoT processes typically list the necessary equipment for curling, such as "curling stone" and "broom", and then check if "curling iron" is on the list. By checking the consistency between CoT and PoT, MoT-1D-Vote is thus able to identify the incorrect or untrustworthy answer.
>
>
>
> It is worth noting that, judging from the results of Figure 9, PoT is not better than CoT, but the combination of CoT and PoT generates diverse thoughts and answers, instead of leaning towards one kind of thinking, thus reducing errors in factual reasoning. Therefore, we can still leverage consistency checking across MoT prompts in decision-making to check if the answer from the weaker LLM is trustworthy in factual-based reasoning tasks. For more details, please refer to Appendix J.
>
> We hope that this experiment can address your concern and once again demonstrate the effectiveness of our method on reasoning tasks. If you have further questions or comments, please let us know and we will try to address them during the remaining rebuttal period!

---

### Official Review · Reviewer_BfsR · 2023-11-01

**Soundness:** 4 excellent
**Presentation:** 3 good
**Contribution:** 3 good
**Rating:** 6
**Confidence:** 3

**Summary:**

This paper addresses the problem of robust and cost-efficient question answering using LLMs. To reduce the cost of accurate question answering, the paper proposes to estimate a weak LLM's uncertainty about its answer, to decide whether to accept the answer or reject it and instead ask a strong (but more expensive) LLM. The paper comprehensively evaluates 10 different approaches to the "routing" task, and compares the proposed approach to several baselines. The experiments show that significant cost savings are possible without compromising on task accuracy (relative to always using the strong LLM).

**Strengths:**

I found the paper to be of high quality and clarity. Specific strengths include:

* This is a clearly written paper that proposes and comprehensively evaluates a simple technique for reducing the cost of question answering with language models.

* The empirical evaluation is impressively thorough, comparing to many interesting baselines.

* The paper surfaces several interesting ideas, e.g., that the sampling distribution of an LLM alone may be insufficient for evaluating how uncertain it is, but by varying the prompting strategy, it is possible to get a broader distribution over LLM answers (which may more accurately reflect the LLM's uncertainty over the correct answer).

* The paper does not overclaim: it honestly represents itself as a careful empirical study of the value of a particular approach to rational cost-aware decision making in the LLM Q&A setting, and does not overstate its novelty w.r.t. related work.

**Weaknesses:**

1. The evaluation reports "end-to-end" accuracy of the entire cascade under different experimental settings, but does not perform a finer-grained analysis of a key novel component: the uncertainty quantification via sampling. It would be great to see some form of **calibration analysis**: in the vote-based methods, how calibrated is the distribution over sampled answers? That is, for each number 1 <= n <= K, how often are the answers that receive n votes actually correct answers? In a perfectly calibrated model, n/K of the answers receiving n votes (across the entire dataset) would be correct answers. Even without perfect calibration, it is interesting to see if the calibration plot is at least monotone: do answers that receive more votes have a higher probability of being correct? It would be great to see how calibration varies across the various vote-based sampling procedures, and perhaps across different LLM temperatures.

Such analyses would contribute new evidence on important scientific questions surrounding language models, like the extent to which LLMs "know what they don't know", and how this uncertainty can best be quantified. For example, the paper
https://arxiv.org/pdf/2207.05221.pdf reports that explicitly asking an LLM to evaluate the truthfulness of a proposed answer yields a calibrated distribution over the tokens True and False. Does the present paper's "External Verifier - QA" setting provide contrary evidence? To evaluate this, it would be helpful to see the calibration of the External Verifier compared to the calibration of the methods this paper proposes. (Also, it would likely be necessary to set the temperature higher than 0.4 -- the other paper reports calibration for temperature 1.0 for base language models, and temperature 2.5 for RLHF-tuned models.)

2. Cost is measured based on the actual cost of using GPT-3.5 and GPT-4. This is not unreasonable (and is the exact calculation that many potential users of this framework might wish to do), but the lack of transparency around OpenAI's pricing model, and how it relates to the actual costs of running strong and weak models, makes it harder to interpret the paper's results. I don't think it's necessary for acceptance, but it would be nice to see whether the results from the paper still hold up when using e.g. Llama 2-7b vs. 70b variants, for some replicable measure of cost.

**Questions:**

* How exactly does the MoT-2D setting work? There are now four prompts, rather than two. In the voting setting, this poses no additional problems, but what about the verification setting? Do all four prompt settings have to agree? Or are multiple prompts "pooled" when computing two vote-based answers to compare for verification?

* If temperature 0.8 yields better results (Fig. 5), why is this not your default? Did you try increasing the temperature further (e.g. Temperature 1)?

* In Figure 4, what threshold was used to decide whether answers were consistent or not?

* Why do you think QA-based external verification with GPT-3.5 performed poorly? Does it incorrectly validate many incorrect answers as trustworthy? Have you tried increasing the temperature of the QA-based verifier, to understand the actual distribution the model places on "yes, trustworthy" vs. "no, not trustworthy"?

---

> ### Author Response · Authors · 2023-11-20
> **Thank you for your review**
>
> Thank you for recognizing our work being of high quality and clarity and for providing us with insightful suggestions!
>
> We have revised our manuscript and addressed all the points you mentioned:
>
> **W1: In the vote-based methods, how calibrated is the distribution over sampled answers?**
>
> Thanks for your feedback! We agree with your suggestion that fine grained calibration analysis is important. In Section 3.3 and Figure 4, we have tried to have more in-depth analysis on the consistency or agreement rate of different vote-based approaches. In our updated Appendix I, we further include a calibration analysis as you suggested. In our analysis, we compare MoT-1D-Vote, CoT-1D-Vote, CoT-2D-Vote, with two variants of LLM-QA following the design of Kadavath et al. (2022), employing T=1 and T=2. Detailed results of this experiment are presented in Figure 8 (left).
>
>
>
> Our analysis indicates that all decision-making methods yield a monotone calibration curve, implying that when they have higher confidence in a certain answer, the answer is generally more likely to be true. But there is no a significant difference among these approaches in terms of their calibration degree.
>
>
>
> However, we wanted to note that achieving perfect calibration with $n/K$ as the confidence score is not necessary for our task. A more direct comparison of the accuracy of the subset satisfying $n/K$ greater than the confidence score among different decision-making approaches is in Figure 8(right). We observe the subset accuracy increases monotonically with a larger $n$ and our method is better than the LLM-QA method, which explains its superiority in the main experiments. For more details and discussions, please refer to Appendix I.
>
> **W2: The lack of transparency around OpenAI's pricing model.**
>
> We agree with your concerns regarding the lack of transparency around OpenAI's pricing model, which may make our results hard to interpret. In our experiments, we have assumed that the monetary cost difference between GPT-4 and GPT-3.5-turbo can reflect the difference in their computational costs, hoping that our results could still provide insights for the latter case. Transferring our results to the cost model of LLAMA2 could be tricky because LLAMA2 can have very different performance compared with GPTs (see Section 3.6 for our effort on this aspect).
>
> To increase the transparency under this restricted condition, we instead decided to release all inputs and outputs from our experiments, as well as a Python script for calculating the token counts for each decision maker on the dataset. We uploaded our demo script with GSM8k dataset and will release the script for all datasets in the future. In this way, researchers or developers in the future could play with our token counts under any monetary or computational cost measurement they prefer. While this may not completely resolve your concern, we hope our effort can still contribute towards a more transparent use of our approaches.

---

> ### Author Response · Authors · 2023-11-20
> **Thank you for your review -- Cont.**
>
> **Q1: How exactly does the MoT-2D setting work?**
>
> We apologize for the confusion. For MoT-2D, we sample from two prompts, i.e., one prompt including M shots of examples written in CoT, and another prompt including another M shots of examples written in PoT. The verification score can then be calculated using the outcomes of these two prompts following Eq (3). To eliminate randomness caused by the pairing between demonstration examples and representations, in our experiments, we reported an average result of two cross-pairing. That is, we experimented with CoT1, CoT2, PoT1, and PoT2, where CoT1 and CoT2 denote prompts based on two sets (Set 1 and Set 2) of demonstration examples, but all written in CoT, respectively, and PoT1 and PoT2 similarly denote prompts based on Set 1 and Set 2 of demonstration examples but all written in PoT, respectively. The reported result is an average of pairing CoT1 with PoT2, and pairing CoT2 with PoT1.
>
>
>
> **Q2: If temperature 0.8 yields better results (Fig. 5), why is this not your default? Did you try to increase the temperature further?**
>
> No, we didn’t try to tune the temperature. The setting of T = 0.4 is from the well-known source paper (https://arxiv.org/abs/2211.12588). We borrowed their experience and intentionally kept the same temperature in our initial experiments.  In the robustness analysis, although we found that we can get a higher accuracy with T = 0.8, re-running all experiments with a different temperature can consume a lot of money and time. Therefore, we have not repeated the experiments with a new temperature. We did not try to increase the temperature further, but this could be an interesting investigation in the future.
>
>
>
> **Q3: In Figure 4, what threshold was used to decide whether answers were consistent or not?**
>
> We apologize for the confusion when describing our Figure 4! The “consistency rate” mentioned in the Y-axis of Fig 4 refers to the same consistency or agreement score in our Eq (2). Below, we clarify how we collected the statistics in Fig 4, and these details have been clarified in our revised draft:
>
> For each vote-based decision-making method, we first group questions into “easy” and “hard” based on whether the weaker LLM can answer them correctly (i.e., whether the majority-voted answer $A^w$ is correct or not).   For each answer, we then calculate its consistency/agreement score following Eq (2).  The Y-axis of Fig 4 reports an average consistency score across all easy (blue bar) or hard (green bar) questions.
>
> Our results in Fig 4 imply that MoT-Vote outperforms CoT-Vote and PoT-Vote because it often assigns relatively lower consistency scores to hard questions while relatively higher ones to easy questions. As a result, when setting up the vote-based threshold, it is more successful in identifying hard questions (what weaker LLM cannot solve) and passing them to the stronger LLM, while saving costs by keeping the easy questions (what weaker LLM can solve).
>
>
>
> **Q4: Why do you think QA-based external verification with GPT-3.5 performed poorly? Does it incorrectly validate many incorrect answers as trustworthy? Have you tried increasing the temperature of the QA-based verifier, to understand the actual distribution the model places on "yes, trustworthy" vs. "no, not trustworthy"?**
>
> We gave our analysis at the end of section 3.5, that is, "It's an intrinsic challenge of deciding question difficulty and answer correctness solely based on their textual descriptions." Similar conclusions are also mentioned in some other papers (https:// arxiv.org/pdf/2303.17651.pdf, https://arxiv.org/abs/2306.13063).
>
> From Figure 8, we could learn that the decision maker precision of the QA-based external verification is lower than our approach, indicating that many untrustworthy answers are trusted incorrectly (i.e., false positive). We increased the temperature in Appendix I and found that it could help the LLM-QA method but is still worse than our approach.

---

> > ### Comment · Reviewer_BfsR · 2023-11-23
> >
> > Thanks for your responses! I appreciate the new plots showing calibration.
> >
> > I am still a bit confused by MoT-2D: it sounds like you just averaged the results from two different experiments with MoT-1D (but with different prompts)? Does this mean that obtaining MoT-2D results uses twice as much compute as the other methods? Please clarify in the revised paper.
> >
> > I am not changing my score and still support the acceptance of this paper.

---

### Official Review · Reviewer_ijvt · 2023-11-07

**Soundness:** 4 excellent
**Presentation:** 4 excellent
**Contribution:** 4 excellent
**Rating:** 8
**Confidence:** 4

**Summary:**

This work introduces an interesting cascading approach to reduce cost in LLM inference. They devised a method: for easy questions, it will use a cheaper LLM (GPT-3.5). But for really hard questions, we'll use the expensive, stronger LLM (GPT-4). This method consists of a weaker LLM, a stronger LLM, and a decision maker, and they reduced the cost to 40% of the cost in using a stronger LLM for everything. To decide which LLM to use, they check if the simpler version gives consistent answers every time they consider its answer. If it does, the question is probably easy, and they stick with the weaker LLM. But if the answers are all over the place, it means the question is tough, and they switch to a stronger LLM. They tried 10 different strategies using Chain of Thought, Program of Thought, mixture of Thought along with majority vote, and verification-based decision making.to find the optimal way to reduce cost while ensuring equal or better performance.

**Strengths:**

* They used unique ways of prompting for better decision-making, especially sampling from different in-context demonstrations and thought representations.
* In-depth analysis of which strategy worked better and why. Evaluating consistency, robustness, and comparisons to other fine-tuned models gives a deeper understanding of how LLMs work.

**Weaknesses:**

I haven't found any major weaknesses

**Questions:**

* Instead of just the answer as a hint, what if we give the entire CoT or PoT from one of the prompts as a hint? Will that help?
* what if we ask multiple questions at once? won't we reduce the cost more?. (2/3 Questions with context as prompt)

---

> ### Author Response · Authors · 2023-11-20
> **Thank you for your review**
>
> Thank you for your valuable comments and the endorsing of our methods and experiments.
>
> We have addressed the questions below. If you have further questions or comments on our response, please don’t hesitate to let us know!
>
> **Q1: Instead of just the answer as a hint, what if we give the entire CoT or PoT from one of the prompts as a hint? Will that help?**
>
> Thank you for your suggestion! Yes, how to better utilize the information from the weaker LLM in prediction is important. We conducted the experiment with the entire CoT and PoT in the GSM8k dataset as hints. In the previous setting, we use the sentence in the prompt: "Hints: The answer may be close to {CoT Answer} or {PoT Answer}". In the new experiment, we replace it with "Hint 1: {Entire CoT Process} Hint 2: {Entire PoT Process}". We conducted the experiment with the examples that CoT and PoT don't have the same answer. The performance for GPT-4 with the entire CoT and PoT is 0.851, which is lower than in the previous setting (0.867 in Table 12).
>
> We have found that leveraging the entire CoT and PoT cannot yield an improvement in performance. Moreover, this approach incurs significant additional costs by necessitating the inclusion of the entire thought process in all demonstration examples, contradicting our primary objective of cost efficiency.
>
>
>
> **Q2: What if we ask multiple questions at once? won't we reduce the cost more? (2/3 Questions with context as prompt)**
>
> This is a good suggestion! To answer this question, we have followed the “batch prompting” setup of Cheng et al. (2023), where we grouped a batch of 4 test questions into each API call of the weaker LLM, in addition to the original 8-shot demonstrations. Like in our previous experiments, we obtain multiple samples from running the weaker LLM, and the verification-based method (Eq 3) can then be leveraged independently for each test question. If the answer is rejected by the decision maker, we then feed the rejected cases into the stronger LLM. We observed adding batch prompting can further reduce costs but slightly compromise accuracy. More details can be found in Appendix H of our updated draft.

---

> > ### Comment · Reviewer_ijvt · 2023-11-22
> >
> > Thank you, Authors, for answering my queries. There is no change in my rating.

---

> > > ### Author Response · Authors · 2023-11-22
> > > **Thank you!**
> > >
> > > Dear reviewer, we are glad to know that we answered your questions well. Thank you again for your recognition of our work!

---

### Author Response · Authors · 2023-11-20
**Summary response to all reviewers and the new revision**

We appreciate all the reviewers for their valuable feedback and constructive suggestions. It is gratifying to note that our work has been well-received, with the reviewers acknowledging its clarity, interest, and comprehension of the experiments and analysis.

In response to the insightful comments from the reviewers, we have revised and updated our manuscript. These revisions are marked in blue in the updated PDF version. Key updates include:

1. Incorporated a new Appendix H that discusses the conjecture in relation to batch prompting.
2. Revised the present statement for Figure 4.
3. Introduced a calibration analysis in Appendix I that examines how our decision makers and the text-based verifiers are calibrated to reflect the true accuracy of their decision making.
4. Revised the limitations section to highlight the potential issue of overconfidence.
5. Included a new Appendix J that presents an experiment and analysis focused on a fact-based question-answering task.

If you have further questions or comments on our submission, please don’t hesitate to let us know. We are happy to have more discussions in the remaining rebuttal period. Thank you!

---

### Comment · Area_Chair_cMG5 · 2023-11-22
**less than one day**

Dear Reviewers,

If you have already responded to authors last response, Thank you!
If not, please take some time, read their responses and acknowledge by replying to the comment. Please also update your score, if applicable.

Thanks everyone for a fruitful, constructive, and respectful review process.

Cheers, Your AC!

---

### Meta-Review · Area_Chair_cMG5 · 2023-12-10

**Metareview:**

This paper proposes using a cascade of weak and strong LLMs to reduce the cost of reasoning tasks by not forwarding all questions to the strong model. For easy question it uses a weaker and cheaper model, and for hard questions, it switches to a stronger, more expensive model. It proposes and evaluates various methods to determine when to switch between the models and shows they can reduce costs by 60% with similar performance to just using the stronger model.

Strengths:
- Tackling a very important problem with a simple and intuitive yet effective solution.
- Comprehensive evaluation of 10 different cascade strategies across 6 reasoning datasets.
- In-depth analysis providing insights into model consistency, robustness, and performance.
- Cost savings of 60% with similar accuracy to only using the expensive model.

Weaknesses:
- The paper doesn't discuss  calibration analysis of the uncertainty quantification method that are applicable to LLMs
- The paper doesn't dive deep into potential limitation of overconfidence on fact-based reasoning tasks.

What may be missing:
- experiments with more transparent pricing model and with various compute costs
- Calibration analysis of the consistency scores for uncertainty quantification.
- Evaluation on fact-based reasoning tasks or retrieval augmented to first study and then maybe address overconfidence limitations.

**Justification For Why Not Higher Score:**

The paper needs more iterations to compare with and employ the recent literature on calibration, over confidence and uncertainty estimation.

**Justification For Why Not Lower Score:**

The paper tackles a very important problem with a very nice and intuitive solution that seems to be working very good. It may offer a lot of insight for future works on calibration and uncertainty estimation especially  for LLMs.

---

### Decision · Program_Chairs · 2024-01-16

Accept (poster)